# Identification of the Effects of Aspirin and Sulindac Sulfide on the Inhibition of HMGA2-Mediated Oncogenic Capacities in Colorectal Cancer

**DOI:** 10.3390/molecules25173826

**Published:** 2020-08-22

**Authors:** Titus Ime Ekanem, Wei-Lun Tsai, Yi-Hsuan Lin, Wan-Qian Tan, Hsin-Yi Chang, Tsui-Chin Huang, Hsin-Yi Chen, Kuen-Haur Lee

**Affiliations:** 1Ph.D. Program for Cancer Molecular Biology and Drug Discovery, College of Medical Science and Technology, Taipei Medical University and Academia Sinica, Taipei 11031, Taiwan; titusekanem@yahoo.com; 2Department of Hematology, University of Uyo, Uyo 520271, Nigeria; 3Graduate Institute of Cancer Biology and Drug Discovery, College of Medical Science and Technology, Taipei Medical University, Taipei 11031, Taiwan; mebar1995@gmail.com (W.-L.T.); amanda_tan@tmu.edu.tw (W.-Q.T.); tsuichin@tmu.edu.tw (T.-C.H.); 4Nutrition and Health Sciences, College of Nutrition, Taipei Medical University, Taipei 11031, Taiwan; ba06106066@tmu.edu.tw; 5Graduate Institute of Metabolism and Obesity Sciences, College of Nutrition, Taipei Medical University, Taipei 11031, Taiwan; hsinyi.chang@tmu.edu.tw; 6Ph.D. Program for Cancer Molecular Biology and Drug Discovery, College of Medical Science and Technology, Taipei Medical University, Taipei 11031, Taiwan; 7TMU Research Center of Cancer Translational Medicine, Taipei Medical University, Taipei 11031, Taiwan; 8Cancer Center, Wan Fang Hospital, Taipei Medical University 11696, Taipei, Taiwan

**Keywords:** colorectal cancer, HMGA2, aspirin, sulindac sulfide, inflammation

## Abstract

Distant metastatic colorectal cancer (CRC) is present in approximately 25% of patients at initial diagnosis, and eventually half of CRC patients will develop metastatic disease. The 5-year survival rate for patients with metastatic CRC is a mere 12.5%; thus, there is an urgent need to investigate the molecular mechanisms of cancer progression in CRC. High expression of human high-mobility group A2 (HMGA2) is related to tumor progression, a poor prognosis, and a poor response to therapy for CRC. Therefore, HMGA2 is an attractive target for cancer therapy. In this study, we identified aspirin and sulindac sulfide as novel potential inhibitors of HMGA2 using a genome-wide mRNA signature-based approach. In addition, aspirin and sulindac sulfide induced cytotoxicity of CRC cells stably expressing HMGA2 by inhibiting cell proliferation and migration. Moreover, a gene set enrichment analysis (GSEA) revealed that gene sets related to inflammation were positively correlated with HMGA2 and that the main molecular function of these genes was categorized as a G-protein-coupled receptor (GPCR) activity event. Collectively, this is the first study to report that aspirin and sulindac sulfide are novel potential inhibitors of HMGA2, which can induce cytotoxicity of CRC cells stably expressing HMGA2 by inhibiting cell proliferation and migration through influencing inflammatory-response genes, the majority of which are involved in GPCR signaling.

## 1. Introduction

Around a quarter of colorectal cancer (CRC) patients are incurable at diagnosis, and half of the patients who undergo potentially curative surgery will ultimately develop metastatic disease [1]. Distant metastatic disease of CRC is present in approximately 25% of patients at initial diagnosis, and half of CRC patients will develop metastatic disease [2]. In the group of patients who develop metastatic disease, the median survival has improved from less than 10 months with the best supportive care to 14 months with fluoropyrimidine treatment [3] and to more than 2 years with a combination of various cytotoxic agents [4]. In spite of advances in systemic therapy, the 5-year survival rate for metastatic CRC is still a mere 12.5% [5]. Therefore, there is an urgent need to investigate the molecular mechanisms of cancer progression in CRC.

High-mobility group A2 (HMGA2) is a non-histone chromatin DNA-binding protein that modulates the transcription of several genes by binding to AT-rich sequences in the minor groove of B-form DNA and altering the chromatin structure [6]. It was demonstrated that overexpression of the HMGA2 protein occurs in various types of cancer, including CRC [7]. It was also shown that high expression of HMGA2 is related to tumor progression, a poor prognosis, and a poor response to therapy [8]. A variety of biological processes are influenced by HMGA2, including the cell cycle process, apoptosis, and the epithelial-to-mesenchymal transition (EMT) [9]. Thus, HMGA2 is an attractive target for cancer therapy.

Bioinformatics methodologies have become a crucial part of drug discovery [10]. This is mostly because they can impact the entire drug development trajectory, by identifying and discovering potential new drugs with a significant reduction in the traditionally high costs and long periods required for new drug development [11]. A scan of publicly available databases containing differential genome-wide messenger (m)RNA signatures identified by researchers in this field can be conducted to signify the perturbation of biological and physiological systems by bioactive small molecules [12]. Our previous study found that the heat shock protein 90 (HSP90) inhibitor has therapeutic potential to inhibit HMGA2-triggered tumorigenesis by this approach [13]. However, 43% of breast cancer patients suffered visual disturbances after drug treatment [14]. Therefore, an urgent need remains to discover and develop potential therapeutic drugs to suppress the overexpression of HMGA2 in cancer patients.

## 2. Results

### 2.1. Aspirin and Sulindac Sulfide as Potential Inhibitors of HMGA2

We first analyzed two Gene Expression Omnibus (GEO) datasets deposited in the National Center for Biotechnology Information (NCBI) databases to discover new inhibitors of HMGA2. In the left panel of Figure 1, gene expression profiles from *HMGA2*-knockdown of human esophageal cancer cells (TE-8) derived from the GEO dataset, GSE143882, are displayed. In the right panel of Figure 1, gene expression profiles from *HMGA2* overexpression of hematopoietic stem and progenitor cells (HSPCs) derived from the GEO dataset, GSE107594, are shown [15]. Furthermore, the top 100 differentially expressed genes (DEGs) from the two GEO datasets were analyzed and queried using the Library of Integrated Network-Based Cellular Signatures (LINCS) L1000 platform to predict which drugs might have potency to inhibit HMGA2 expression. A negative enrichment score (NES) indicates that the gene signature of a drug is opposite to the gene signature of the disease, while a positive enrichment score (PES) is displayed when they are concordant. Results shown in Appendix A indicate the top 10 chemical perturbagens with a PES for the gene expression signature with *HMGA2*-knockdown. Meanwhile, Appendix A shows the top 10 chemical perturbagens with an NES for gene expression signatures with HMGA2 overexpression. Comparing these two tables, we found that the strongest therapeutic predictors of HMGA2 were aspirin and the aspirin-like compound, sulindac sulfide, which are non-steroidal anti-inflammatory drugs (NSAIDs) (Figure 1).

### 2.2. Aspirin and Sulindac SulfideAttenuate the Proliferation of CRC Cells Stably Expressing HMGA2

To gain insights into the interactions of aspirin and sulindac sulfide with HMGA2, molecular docking was carried out. The AT-hook motif as presented in the amino acid sequence of Pro-Arg-Gly-Arg-Pro is a particular feature of HMGA2 [16]. The chemical structures of aspirin and sulindac sulfide are shown in Figure 2A. We next retrieved the sequence of the AT-hook motif from the Protein Data Bank (PDB) (accession code: 3UXW) [17]. Results obtained from SwissDock provided insights into the most favorable binding site for aspirin (Figure 2B) and sulindac sulfide (Figure 2C) in the pocket of the AT-hook motif of HMGA2. The binding free energies were −7.85 kcal/mol for aspirin and −7.68 kcal/mol for sulindac sulfide. Furthermore, we established stable DLD-1 cell lines which overexpressed either an empty vector (called DLD-1 vector) or HMGA2 (called DLD-1 HMGA2). As shown in Figure 2D, expressions of HMGA2-green fluorescent protein (GFP) and endogenous HMGA2 were characterized using an anti-HMGA2 antibody for Western blotting. Since aspirin and sulindac sulfide were predicted to be novel inhibitors of HMGA2, we wanted to investigate whether DLD-1 cells stably expressing HMGA2 were sensitive to aspirin and sulindac sulfide. Thus, we used aspirin and sulindac sulfide to perform a cell viability assay. As expected, the proliferation index of DLD-1 HMGA2 cells significantly decreased in the aspirin-treated group compared to the DLD-1 vector (*p* = 0.011) (Figure 2E). Similar effects were observed in the case of sulindac sulfide-treated groups (Figure 2F). Taken together, these results suggest that both aspirin and sulindac sulfide can attenuate the growth of CRC cells, especially DLD-1 cells stably expressing HMGA2, through a direct interaction between aspirin or sulindac sulfide and HMGA2.

### 2.3. Aspirin and Sulindac Sulfide Decrease the Migratory Ability of CRC Cells Stably Expressing HMGA2

To further determine the influence of aspirin and sulindac sulfide on the migration of CRC cells stably expressing HMGA2, we first examined the migratory ability of DLD-1 vector and DLD-1 HMGA2 cells using a transwell migration assay. As expected, the migratory ability of DLD-1 HMGA2 cells significantly increased by about 75% compared to that of DLD-1 vector cells (*p* = 0.012) (Figure 3A). We further investigated the effect of HMGA2 on the expressions of EMT effectors in DLD-1 vector and DLD-1 HMGA2 cells. As shown in Figure 3B, HMGA2 overexpression did indeed cause induction of mesenchymal markers (i.e., Twist, Snail, and vimentin), in conjunction with concomitant decreases in E-cadherin expression. The above results indicated that the 50% inhibitory concentration (IC_50_) values of growth inhibition were approximately 2.5 mM for aspirin-treated groups (Figure 2E) and approximately 100 µM for sulindac sulfide-treated groups (Figure 2F). Thus, concentrations of 2.5 mM and 100 µM were respectively selected to determine the effects of aspirin and sulindac sulfide on the migration of DLD-1 vector and DLD-1 HMGA2 cells. As shown in Figure 3C, the number of migrating cells among aspirin- or sulindac sulfide-treated DLD-1 HMGA2 cells was significantly reduced compared to the DLD-1 vector group. Expressions of the EMT effectors were further examined in DLD-1 vector and DLD-1 HMGA2 cells after aspirin and sulindac sulfide treatment for 24 h. Our data show that stable expression of HMGA2 enhanced the effect of aspirin- or sulindac sulfide-mediated suppression of the EMT in DLD-1 HMGA2 cells compared to the control vector group. Taken together, these results suggest that aspirin and sulindac sulfide can inhibit the migratory ability and EMT effector expression of CRC cells, especially in DLD-1 cells with stable expression of HMGA2.

### 2.4. HMGA2 Expression is Related to Inflammatory Response Genes Involved in GPCR Signaling

The above results show that inhibitory effects of aspirin and sulindac sulfide are involved in the stable expression of HMGA2 in CRC cells. To understand whether alterations of inflammatory response genes are related to HMGA2, gene sets regulated by HMGA2 from Figure 1 were analyzed. A gene set enrichment analysis (GSEA) revealed that HMGA2 expression was positively correlated with inflammation signal gene signatures in TE-8 cells (Figure 4A) and HSP cells (Figure 4B). Next, to identify the exact association of inflammatory response genes with HMGA2, 148 downregulated inflammatory-response genes with HMGA2-knockdown from the GSEA and 93 upregulated inflammatory-response genes with HMGA2 overexpression from the GSEA were selected and compared to each other. As shown in Figure 4C, 67 overlapping genes were identified from the intersection between the two gene groups (Appendix A). Furthermore, in order to further characterize the 67 overlapping genes, we performed a gene enrichment analysis on the set of all 67 overlapping genes via a FunRich software analysis [18]. The enrichment results showed that 20.9% of the genes were categorized as having GPCR activity, 14.9% of the genes as having receptor activity, and 7.5% of the genes as having cytokine and chemokine activity, according to a molecular function analysis (Figure 4D). Next, to understand how these 67 overlapping genes interact with HMGA2, a predictive network model was employed using the Search Tool for the Retrieval of Interacting Genes/Proteins (STRING) database [19]. The STRING network of HMGA2 and the 67 overlapping genes is shown in Figure 4E, where the preferred association of the node proteins with HMGA2 was interleukin (IL)-6 (enrichment *p* value of <1.0e-16). Collectively, these results suggested that the main influence of HMGA2 on inflammatory response genes involves GPCR signaling.

## 3. Discussion

Several studies demonstrated that the biological functions of HMGA2 are associated with various behaviors of tumor cells, such as proliferation, invasion, and metastasis; thus, HMGA2 is an attractive target for cancer treatment [8,9]. This is the first study to identify the clinical use of aspirin and sulindac sulfide as novel potential inhibitors of HMGA2. Here, we summarize the evidence that supports this conclusion. First, we identified aspirin and sulindac sulfide as novel potential inhibitors of HMGA2 by analyzing two GEO datasets, including the knockdown and overexpression of *HMGA2* gene expression profiles, using the LINCS L1000 platform. Second, we found that DLD-1 cells stably expressing HMGA2 were sensitive to both aspirin and sulindac sulfide treatment. Third, we found that the migratory ability of CRC cells stably expressing HMGA2 was inhibited by both aspirin and sulindac sulfide treatment through HMGA2-regulated EMT effectors. Fourth, we found that HMGA2 expression was related to inflammatory-response genes involved in GPCR signaling. Collectively, this is the first report to describe aspirin and sulindac sulfide as novel potential inhibitors of HMGA2, which can induce cytotoxicity in CRC cells overexpressing HMGA2 by inhibiting cell proliferation and migration through influencing inflammatory-response genes, the majority of which are involved in GPCR signaling.

HMGA2 was demonstrated to play important roles in promoting cell migration and in regulating the EMT pathway through regulating the transcription of several EMT-related genes in various cancers [9,20]. Expressions of Twist and Snail were demonstrated to be upregulated by HMGA2, and the upregulation of Twist and Snail caused downregulation of E-cadherin expression [21]. Our study showed that HMGA2 overexpression was positively correlated with an increased rate of migration of DLD-1 cells with stable expression of HMGA2, which was consistent with previous findings. We next demonstrated that both aspirin and sulindac sulfide treatment could suppress the migratory ability of CRC cells stably expressing HMGA2 (Figure 3). In addition, HMGA2 overexpression enhanced the effects of aspirin- and sulindac sulfide-mediated suppression of EMT effectors in DLD-1 cells stably expressing HGMA2 compared to the control vector group. This indicates that aspirin and sulindac sulfide treatment might be able to inhibit the migration of CRC cells through inhibiting EMT effectors regulated by HMGA2.

Numerous studies demonstrated that HMGA2 can drive inflammatory pathways, facilitating tumor progression and refractory disease [22]. Inflammatory-response genes include cytokines, cytokine receptors, chemokines, chemokine receptors, and transcription factors [23]. Cytokines and chemokines can bind and activate their receptors, GPCRs, which are imbedded in leukocyte cell membranes, thereby inducing leukocyte adhesion to vessel walls, morphological changes, extravasation into inflamed tissues, and chemotaxis along the chemokine gradient to the site of injury or infection [24]. GPCRs represent the largest family of signaling molecules in the human genome [25]. They are activated by a large variety of stimuli, such as peptides, proteins, lipids, ions, neurotransmitters, and hormones. It was reported that stimulation of GPCRs can engage a broad range of physiological responses, such as cell proliferation and migration, chemotaxis, smooth muscle contraction, and neurotransmission. Thus, GPCRs are important targets for current drugs and drug discovery, largely because of the wide range of physiological and pathophysiological processes on which GPCR targeting can have major impacts [25]. In this study, we found that gene sets related to inflammation were positively correlated with HMGA2, and that the main molecular functions of those genes were categorized as a GPCR activity event, which corresponds to results in Figure 1, and indicates that NSAIDs are novel potential inhibitors of HMGA2. Recently, there has been interest in targeting GPCR-related upstream or downstream molecules rather than the GPCRs directly [26]. This approach has the potential to achieve broad efficacy by blocking pathways shared by GPCRs [27]. Furthermore, it is interesting that as illustrated by STRING (Figure 4E), among the 67 overlapping inflammatory-response genes, IL-6 was predicted to interact with HMGA2. It was demonstrated that HMGA2 overexpression significantly provoked the IL-6 protein level in RAW264.7 cells [20]. In addition, upregulation of the HMGA2/IL-6 axis was shown to be highly correlated with poor outcomes in glioblastoma patients [28]. Thus, HMGA2 might positively regulate the expression of inflammatory-response genes by activating the IL-6 pathway, and this may offer us new insights into HMGA2-mediated regulation of inflammation.

NSAIDs are members of a drug class that reduces pain, decreases fever, prevents blood clots, and in higher doses, decreases inflammation [29]. There are currently at least 20 approved NSAIDs, the most prominent of which are aspirin, ibuprofen, and naproxen, all of which are available over the counter in most countries [30]. The concept that NSAIDs can prevent CRC is now well established, owing to huge epidemiologic studies which indicated that prolonged use of NSAIDs reduced adenomatous polyps, as well as the incidence of disease and death from CRC by 30~50%, and most interventional studies focused on aspirin and sulindac sulfide [31,32]. In addition, it was reported that aspirin can reduce the occurrence of CRC, and also reduce the risk of metastasis in CRC [33]. However, the mechanism of action of aspirin and sulindac sulfide as anticancer agents remains unclear. In this study, we used gene expression profiles of HMGA2-knockdown and overexpression, and employed the LINCS L1000 platform which predicted that aspirin and the aspirin-like compound, sulindac sulfide, are novel inhibitors of HMGA2. We found that the migratory ability of CRC cells with stable expression of HMGA2 was inhibited by aspirin and sulindac treatment through HMGA2-regulated EMT effectors. These findings indicate that aspirin and sulindac sulfide might act as inhibitors of the HMGA2-induced migration of CRC cells, and that HMGA2 could be a potential therapeutic target for HMGA2-induced EMT and metastasis of CRC.

In conclusion, results presented in this study provide strong support for further exploration of aspirin and the aspirin-like compound, sulindac sulfide, as novel potential inhibitors of HMGA2 bythe genome-wide mRNA signatures-based approach. Aspirin- and sulindac sulfide-induced cytotoxicity of CRC cells stably expressing HMGA2 by inhibiting cell proliferation and migration through influencing inflammatory-response genes, the majority of which were involved in GPCR signaling.

## 4. Materials and Methods

### 4.1. Chemicals, Reagents, and Antibodies

Sulindac sulfide, aspirin, methanol, crystal violet, and G418 were obtained from Sigma-Aldrich (St. Louis, MO, USA). Antibodies against green fluorescent protein (GFP), Twist, Snail, vimentin, E-cadherin, α-tubulin, and GAPDH were obtained from Cell Signaling (Beverly, MA, USA). HMGA2 was obtained from Santa Cruz Biotechnology (Santa Cruz, CA, USA).

### 4.2. Prediction of HMGA2 Inhibitors

Computational prediction of novel HMGA2 inhibitors was performed as follows. Differentially expressed gene (DEG) signatures were obtained from the Gene Expression Omnibus (GEO) databases of the National Center for Biotechnology Information (NCBI), with data from HMGA2-knockdown of human esophageal cancer cells (TE-8) (GSE143882) and HMGA2 overexpression of hematopoietic stem and progenitor cells (HSPCs) (GSE107594). Next, signatures of DEGs were analyzed using the Library of Integrated Network-Based Cellular Signatures (LINCS) L1000 platform [34] to predict drugs that have the potency to inhibit HMGA2 expression.

### 4.3. Molecular Docking Analysis

The simulated docking of NSAIDs to the AT-hook motif of HMGA2 was carried out using SwissDock [35] and analyzed using Chimera [36]. The sequence of the AT-hook motif was retrieved from the Protein Data Bank (PDB) (accession code: 3UXW). Binding free energies were averaged to estimate the affinity from the predicted binding intensities of docking interactions of NSAIDs and the AT-hook motif of HMGA2. All docking parameters were set to the default.

### 4.4. Cell Cultures

DLD-1 control vector and DLD-1 HMGA2 stably expressing cell lines were provided by Prof. Huang’s laboratory (Taipei Medical University, Taipei, Taiwan). All CRC cell lines were cultured in RPMI-1640, supplemented with 10% fetal bovine serum (FBS) and antibiotics (all from Thermo Fisher Scientific, Waltham, MA, USA), and maintained at 37 °C in a humidified atmosphere containing 5% CO_2_. The cell lines were selected by 0.5 mg/mL G418.

### 4.5. Cell-Viability Assay

Cell viability was determined using the crystal violet staining method, as described previously [37]. In brief, cells were plated in 96-well plates at 3000 cells/mL and treated with or without NSAIDs at the indicated concentrations. After NSAID treatment for 48 h, cells were stained with 0.5% crystal violet for 10 min at room temperature. Next, the plates were washed with tap water three times. After drying, cells were lysed with a 0.1 M sodium citrate solution (Sigma-Aldrich, St. Louis, MO, USA), and measured at 550 nm on a microplate reader.

### 4.6. Cell-Migration Assay

DLD-1 control and DLD-1 HMGA2 stable expression cell lines (10^4^ cells/well) in 0.5 mL of serum-free medium were seeded in the upper chamber membranes of Falcon^TM^ cell culture inserts (BD Biosciences, San Jose, CA, USA) and received different treatments. These membranes were inserted into 24-well plates containing 10% FBS-supplemented medium. After 24 h, cells were fixed with 100% methanol and stained with 5% Giemsa stain (Merck, Darmstadt, Germany). Non-migrated cells that remained in the upper chambers were removed with a damp cotton swab. Cells that had migrated were photographed under a phase-contrast microscope.

### 4.7. Western Blotting

Cell lines were placed in lysis buffer at 4 °C for 1 h. Protein samples were analyzed using different percentages of sodium dodecylsulfate (SDS)-polyacrylamide gel electrophoresis, as described in [38].

### 4.8. Gene Set Enrichment Analysis (GSEA) and Functional Enrichment Analysis

We used GSEA v4.0 to perform a GSEA on public resources (GSE143882 and GSE107594). Gene sets were obtained either from the molecular signatures database (MSigDB) hallmark gene set [39] or from two public resources (GSE143882 and GSE107594). Statistical significance was assessed by comparing enrichment scores to enrichment results. Significant DEGs were selected for the functional enrichment analysis using the FunRich3.1.3 tool [18].

### 4.9. Statistical Analysis

Results are presented as the mean ± standard deviation (SD). We used Student’s *t*-tests for all comparison experiments. Statistical analyses of the cell-viability and cell-migration assays were performed using an unpaired Student’s *t*-test with Excel software. All *p* values of <0.05 were considered significant.

## Figures and Tables

**Figure 1 molecules-25-03826-f001:**
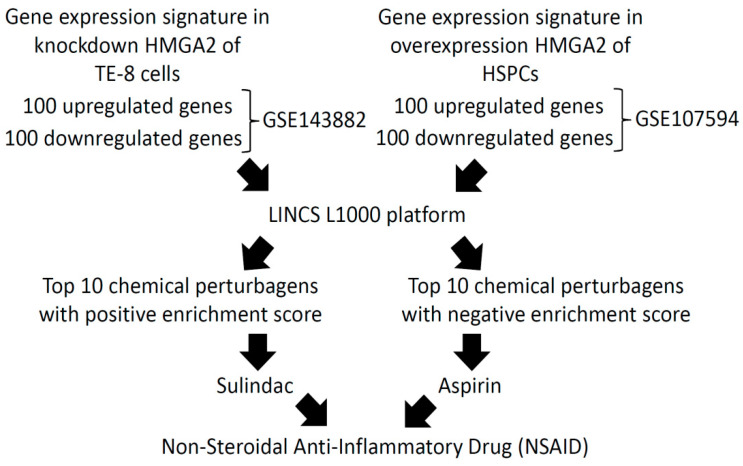
Diagram of new human high-mobility group A2 (HMGA2) inhibitor identification. Two Gene Expression Omnibus (GEO) databases (i.e., GSE143882 and GSE107594) were selected from the National Center for Biotechnology Information (NCBI). The top 100 up- and downregulated genes were analyzed from the two GEO databases and further queried using the Library of Integrated Network-based Cellular Signatures (LINCS) L1000 platform to predict new inhibitors of HMGA2. TE-8, esophageal cancer cells; HSPCs, hematopoietic stem and progenitor cells.

**Figure 2 molecules-25-03826-f002:**
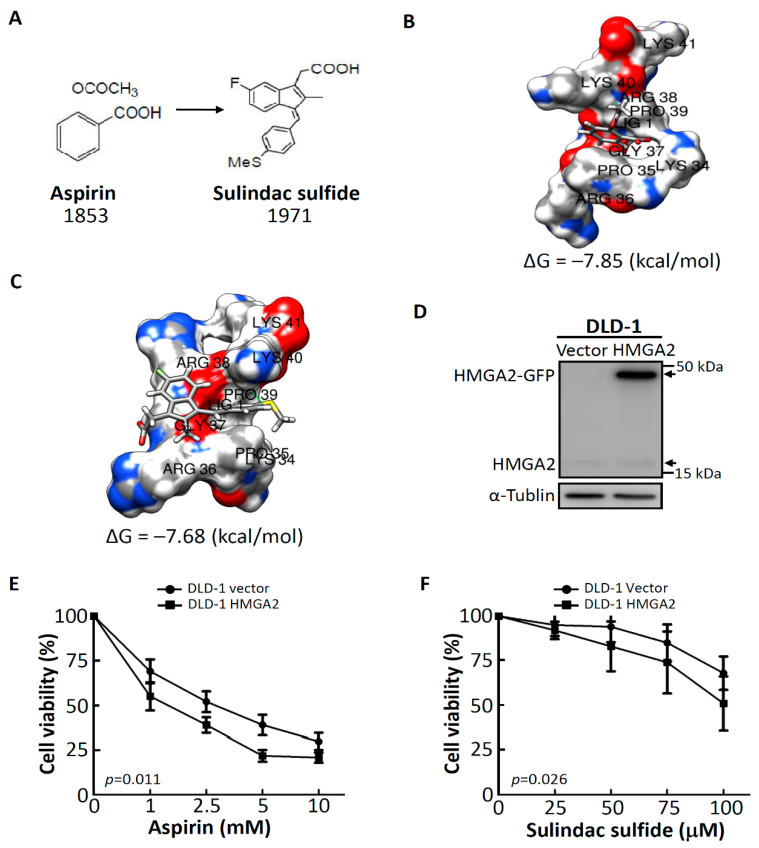
Growth inhibitory effects of aspirin and sulindac sulfide in DLD-1 (empty vector) vector and DLD-1 HMGA2 cells. (**A**) Chemical structures of aspirin and sulindac sulfide. The binding mode of aspirin (**B**) or sulindac sulfide (**C**) fit into the pocket of the AT-hook motif of HMGA2. (**D**) Western blotting with an anti-HMGA2 antibody to examine protein expression levels of HMGA2 in DLD-1 stable cell lines. Cells were treated with aspirin (**E**) or sulindac sulfide (**F**) at the indicated concentrations for 48 h, and cell viability was assessed by the crystal violet staining method. Bars = standard deviation (SD) (*n* = 6).

**Figure 3 molecules-25-03826-f003:**
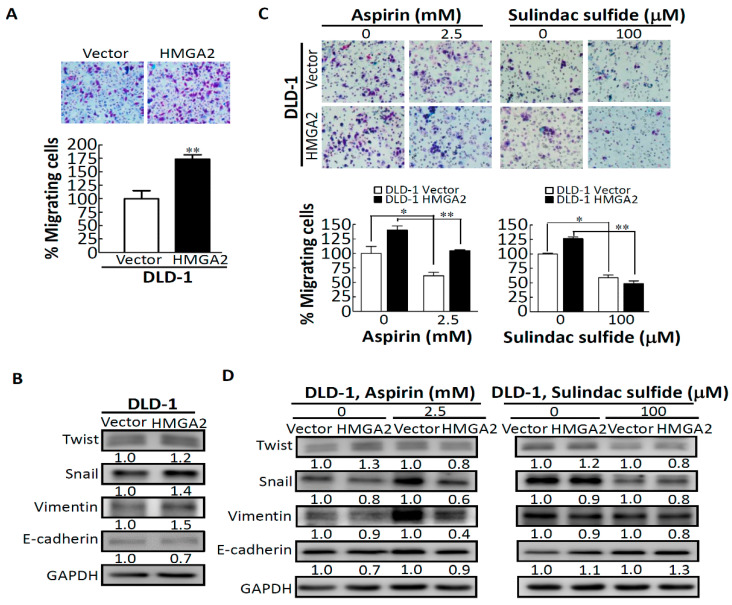
Aspirin- and sulindac sulfide-mediated suppression of the migratory ability and epithelial-mesenchymal transition (EMT) of DLD-1 (empty vector) vector and DLD-1 HMGA2 cells. (**A**) The migratory ability of DLD-1 vector and DLD-1 HMGA2 cells was assessed using a migration transwell assay. (**B**) Protein expressions of EMT effectors in DLD-1 vector and DLD-1 HMGA2 cells, as revealed by gains in the mesenchymal markers, Twist, Snail, and vimentin, and loss of the epithelial marker, E-cadherin. (**C**) Effects of aspirin and sulindac sulfide on the migratory activities of DLD-1 vector and DLD-1 HMGA2 cells after 24 h of treatment. (**D**)Effects of aspirin- or sulindac sulfide-mediated reversal of the mesenchymal character of DLD-1 HMGA2 cells. * *p* < 0.05, ** *p* < 0.01.

**Figure 4 molecules-25-03826-f004:**
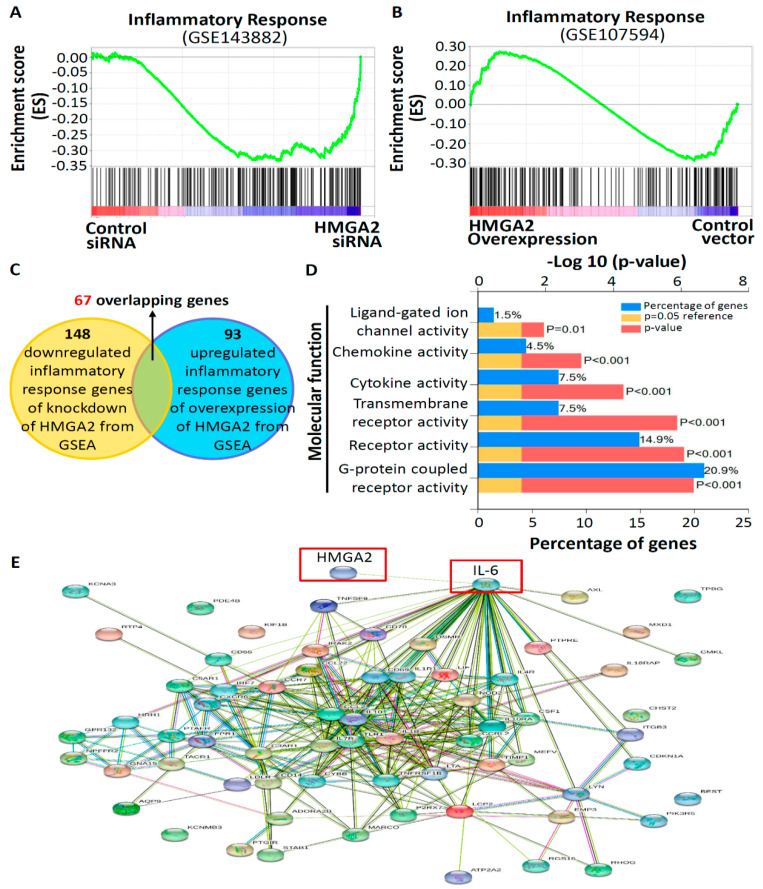
Expressions of inflammatory response genes related to the HMGA2 expression status. Gene set enrichment analysis (GSEA) showed enrichment of a downregulated inflammatory response gene set in TE-8 cells after transfection with HMGA2 siRNA (**A**), or an upregulated inflammatory response gene set in HSPCs after transfection with the HMGA2 plasmid (**B**). (**C**) Sixty-seven overlapping inflammatory response genes were identified from the intersection between 148 downregulated inflammatory-response genes after HMGA2-knockdown and 93 upregulated inflammatory-response genes after HMGA2 overexpression. (**D**) Functional gene enrichment analysis of the molecular functions of 67 overlapping inflammatory-response genes using FunRich software. Blue bar: percentage of genes assigned to the indicated term; yellow bar: reference *p* value; red bar: calculated *p* value of enrichment of the indicated term. (**E**) Search Tool for the Retrieval of Interacting Genes/Proteins (STRING) gene networks of the interactions ofHMGA2 and the 67 overlapping inflammatory response genes. The node protein of potential importance for HMGA2 was interleukin (IL)-6.

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
