# Peer review of "Identification of the Effects of Aspirin and Sulindac Sulfide on the Inhibition of HMGA2-Mediated Oncogenic Capacities in Colorectal Cancer"

_molecules, 2020, doi:10.3390/molecules25173826_

Round 1

Reviewer 1 Report

I appreciate the comments of the authors and I believe that the paper is now acceptable for publication.

Author Response

Response 1: Thanks for reviewer's positive comments.

Reviewer 2 Report

The manuscript “In Silico Identification of the Effects of Non-Steroidal Anti-Inflammatory Drugs on the Inhibition of HMGA2-Mediated Oncogenic Capacities in Colorectal Cancer” is a resubmission of a draft evaluated some weeks ago with title In Silico Identification of Non-Steroidal Anti-Inflammatory Drugs Effect in Inhibition of HMGA2-Mediated Oncogenic Capacities in Colorectal Cancer

In that moment this referee suggested that the work was not adequate for the readers of Molecules due to the mainly bioinformatic approach used by the authors in their identification of potential inhibitors of HMGA2 and the complete absence of chemical insights.

Reading the new version of the draft and considering the little changes made in scientific terms, the opinion of the referee is not changed and suggest again that the work could be more appropriate for a specialized bioinformatic or biological journal.

This referee regrets to suggest again that the work be rejected for publication in Molecules.

Author Response

Response 1: Thanks for reviewer's valuable comments. We submit this manuscript to Special Issue "Anticancer Drug Discovery and Development" of this journal. We provide our experiences and findings on the discovery and development of new therapeutic agents (aspirin and aspirin-like compound, sulindac sulfide) to target on high expression of human high-mobility group A2 (HMGA2) for colorectal cancer (CRC). In addition, we found that NSAIDs induced cytotoxicity of CRC cells stably expressing HMGA2 by inhibition of cell proliferation and migration through direct interaction with the AT-hook motif (a small DNA-binding protein motif) of HMGA2 (Revised Figure 2B and 2C, shown below). Moreover, to understand the level of HMGA2 gene expression in various colorectal cancer (CRC) cell lines, 10 CRC cell lines commonly used in research were selected from CellExpress database [1]. We found that HMGA2 was relative low expressed in DLD-1 cell line among the 10 CRC cell lines, but not completely non-expressive of HMGA2 (Figure A). In addition, the protein expression level of HMGA2 in DLD-1 cells has been demonstrated in previous study [2]. In our study, since aspirin and sulindac sulfide were predicted as a new inhibitor of HMGA2, we established the stable DLD-1 cell lines which overexpressed an empty vector or HMGA2 to investigate whether DLD-1 cells stably expressing HMGA2 were sensitive to NSAIDs. Moreover, we also used HCT116 cell line which with high endogenous HMGA2 expression level [2] to investigate the drug sensitive of aspirin and sulindac sulfide. As shown in below Figure B and C, we found that HCT116 cells were sensitive to aspirin and sulindac sulfide treatment, these results are similar with aspirin and sulindac sulfide-treatedDLD-1 cells stably expressing HMGA2. Finally, to make the title more consistent with the body of the manuscript, the title has changed to “Identification the Effects of Aspirin and Sulindac Sulfide on the Inhibition of HMGA2-Mediated Oncogenic Capacities in Colorectal Cancer”. We have made the big change as above-mentioned, we hope that such a change will arouse audiences’ favor.

Reference

[1] Lee YF, Lee CY, Lai LC, Tsai MH, Lu TP, Chuang EY. CellExpress: a comprehensive microarray-based cancer cell line and clinical sample gene expression analysis online system. Database (Oxford). 2018;2018:bax101.

[2] Leung SW, Chou CJ, Huang TC, Yang PM. An Integrated Bioinformatics Analysis Repurposes an Antihelminthic Drug Niclosamide for Treating HMGA2-Overexpressing Human Colorectal Cancer. Cancers (Basel). 2019; 11(10):1482.

Reviewer 3 Report

I decided to review the manuscript because of the title  - ‘In silico identification of the effects of … ‘. In Abstract is the information about ‘a gene set enrichment analysis … ‘ but the title is misleading.

In silico part is very poor, and it only justify the selection of two drugs to the analysis.  It means that the authors should change the title and the body of the manuscript.

They analyzed two NSAIDs with two different scores, but they did not compare results for them and in Discussion they used a term ‘NSAIDs’ but they analyzed only aspirin and sulindac-sulfide.

However, a search for new inhibitors of HMGA2, and an attempt to explain its mode of action in cancer cells is worth to be publish after rewriting of the manuscript.

I should mention that I am not a specialist in  a gene analysis

Author Response

Response 1: We are sorry for the misleading and thank reviewer's valuable suggestion. We have made the change of the problems as mentioned by reviewer as shown below:

1) In Abstract is the information about ‘a gene set enrichment analysis … ‘ but the title is misleading

Answer: A gene set enrichment analysis shown in abstract to indicate the result of Figure 4 which not related with title.

2) In silico part is very poor, and it only justify the selection of two drugs to the analysis. It means that the authors should change the title and the body of the manuscript.

Answer: We are sorry for the misleading and thank reviewer's valuable suggestion. We have made the change of the title “Identification the Effects of Aspirin and Sulindac Sulfide on the Inhibition of HMGA2-Mediated Oncogenic Capacities in Colorectal Cancer”. In addition, to make the title more consistent with the body of the manuscript, we have replaced “NSAID” to “Aspirin and Sulindac Sulfide” in the body of the manuscript, the detail shown below:

Line 33: In addition, aspirin and sulindac sulfide induced cytotoxicity of CRC cells stably expressing HMGA2 by inhibition of cell proliferation and migration.

Line 38: Collectively, this is the first study to report that aspirin and sulindac sulfide are a novel potential inhibitor of HMGA2

Line 74: Aspirin and Sulindac Sulfide as Potential Inhibitors of HMGA2

Line 90: We found that the strongest therapeutic predictors of HMGA2 are aspirin and aspirin-like compound, sulindac sulfide

Line 93: Aspirin and Sulindac Sulfide Attenuate the Proliferative of CRC Cells Stably Expressing HMGA2

Line 94: To gain the insight into the interactions of aspirin and sulindac sulfide with HMGA2

Line 110: these results suggest that aspirin and sulindac sulfide can attenuate the growth of CRC cells, especially DLD-1 cells stably expressing HMGA2 through direct interaction between aspirin or sulindac sulfide and HMGA2

Line 123: Aspirin and Sulindac Sulfide Decrease the Migratory Ability of CRC Cells Stably Expressing HMGA2

Line 124: To further determine the influence of aspirin and sulindac sulfide on the migration of CRC cells stably expressing HMGA2

Line 142: These results suggest that aspirin and sulindac sulfide can inhibit the migration ability and EMT effector expression of CRC cells, especially in DLD-1 cells with stable expression of HMGA2

Line

Line 269: In addition, the migration ability of the stable expression of HMGA2 in CRC cells was inhibited by aspirin and sulindac sulfide treatment through HMGA2-regulated EMT effectors.

Line 273: In conclusion, the results presented in this study provide strong support for the exploration of aspirin and aspirin-like compound, sulindac sulfide as novel potential inhibitors of HMGA2 by the genome-wide mRNA signatures based approach. Aspirin or sulindac sulfide induced cytotoxicity of CRC cells stably expressing HMGA2 by inhibiting of cell proliferation and migration through influencing the inflammatory response genes, the majority of which were involved in GPCR signaling.

Round 2

Reviewer 2 Report

The manuscript "Identification the Effects of Aspirin and Sulindac Sulfide on the Inhibition of HMGA2-Mediated Oncogenic Capacities in Colorectal Cancer" is the revised version of “In Silico Identification of the Effects of Non-Steroidal Anti-Inflammatory Drugs on the Inhibition of HMGA2-Mediated Oncogenic Capacities in Colorectal Cancer”

This referee suggested that the work was not adequate for the readers of Molecules due to the mainly bioinformatic approach used by the authors in their identification of potential inhibitors of HMGA2 and the complete absence of chemical insights.

Reading the new version of the draft and considering the little changes made in scientific terms (basically only the title has been changed), the opinion of the referee is still the same and suggest again that the work could be more appropriate for a specialized biological journal. Moreover, from a methodological poin of view, the modelling approach (docking calculations) is not relvant considering that no analysis or rationalization is proposed by the authors.

This referee regrets to suggest again to rejected this work for publication in Molecules.

Author Response

We appreciate the comments of this reviewer and believe that our manuscript has been improved by attention to him or her. The followings are our responses to the specific issues raised by this reviewer:

Point 1: The manuscript "Identification the Effects of Aspirin and Sulindac Sulfide on the Inhibition of HMGA2-Mediated Oncogenic Capacities in Colorectal Cancer" is the revised version of “In Silico Identification of the Effects of Non-Steroidal Anti-Inflammatory Drugs on the Inhibition of HMGA2-Mediated Oncogenic Capacities in Colorectal Cancer”. This referee suggested that the work was not adequate for the readers of Molecules due to the mainly bioinformatic approach used by the authors in their identification of potential inhibitors of HMGA2 and the complete absence of chemical insights. Reading the new version of the draft and considering the little changes made in scientific terms (basically only the title has been changed), the opinion of the referee is still the same and suggest again that the work could be more appropriate for a specialized biological journal. Moreover, from a methodological poin of view, the modelling approach (docking calculations) is not relvant considering that no analysis or rationalization is proposed by the authors. This referee regrets to suggest again to rejected this work for publication in Molecules.

Response 1: Thanks for reviewer's valuable comments. For the modelling approach, it is well-known that The AT-hook motif of HMGA2 plays a crucial role in regulating gene expression by binding to minor grooves of DNA at special AT-rich DNA sequences. In order to gain insights into the putative binding mode of aspirin or sulindac sulfide with the AT-hook motif of HMGA2, blind docking for aspirin or sulindac sulfide with the crystallographic model of the AT-hook motif was carried out. The minus binding free energies (−7.85 kcal/mol for aspirin and −7.68 kcal/mol for sulindac sulfide) indicated the binding intensities of the docking interaction of aspirin or sulindac sulfide and the AT-hook motif of HMGA2. Taken together, we do our best to respond the comments of reviewer, and we still hope that our responses can be favored by reviewer.

Reviewer 3 Report

The author treated my comments very  formally. They should comment why they selected two compounds from two different evaluation ways? What they expected to explain/test?

The inhibition effect is observed at high IC50 values of compounds. Toxicity parameters should be also discussed. 

Author Response

We appreciate the comments of this reviewer and believe that our manuscript has been improved by attention to him or her. The followings are our responses to the specific issues raised by this reviewer:

Point 1: The author treated my comments very formally. They should comment why they selected two compounds from two different evaluation ways? What they expected to explain/test?

Response 1: Thanks for reviewer's valuable comments. To identify the novel inhibitor of HMGA2, we analyzed two GEO datasets including HMGA2-knockdown and overexpression datasets and queried to LINCS L1000 platform. to predict which drugs might have potency to inhibit HMGA2 expression. Using these two databases with opposite HMGA2 gene expressions, we found that the drugs predicted to inhibit HMGA2 are aspirin and aspirin-like compound, sulindac sulfide, which are non-steroidal anti-inflammatory drugs (NSAIDs).

Point 2: The inhibition effect is observed at high IC50 values of compounds. Toxicity parameters should be also discussed.

Response 2: Thanks for reviewer's valuable comments. For the toxicity of aspirin, it has been demonstrated that adult deaths occur in patients whose concentrations exceed 700 mg/L (5.1 mM) of aspirin. The concentration of aspirin in our study is only 2.5 mM, which will not cause toxicity problems. In clinical studies, sulindac is about 10-fold to be comparable in effectiveness to aspirin. It is meant that adult deaths occur in patients whose concentrations exceed 70 mg/L (0.51 mM) of sulindac. The concentration of sulindac in our study is only 0.1 mM, which will not cause toxicity problems.

This manuscript is a resubmission of an earlier submission. The following is a list of the peer review reports and author responses from that submission.

Round 1

Reviewer 1 Report

The manuscript “In Silico Identification of Non-Steroidal Anti-Inflammatory Drugs Effect in Inhibition of HMGA2-Mediated Oncogenic Capacities in Colorectal Cancer” by Ekanem et al. presents a bioinformatic in silico identification of potential inhibitors of HMGA2.

Among the best 20 potential drugs identified the authors selected 2, namely Sulinidac sulfide and Aspirin that have been used in subsequent cell migration and viability assays.

The manuscript is in general well written but in opinion of this referee it is not of general interest for the readers of Molecules. The work could be more appropriate for a specialized bioinformatic or biological journal and for this reason suggest that the work should not be accepted for publication in this Journal.

Reviewer 2 Report

In this paper the authors try to demonstrate that two Non-Steroidal Anti-Inflammatory Drugs, namely Aspirin and Sulindac Sulfide may act as inhibitors of HMGA2 protein, in the view of a possible use in CRC therapy.
In my opinion the results presented in this paper do not give convincing evidence to support the hypothesis of the authors.
I am quite puzzled by the results presented in fig. 2 C and D: why the cell overexpressing HMGA2 are more sensitive to the inhibitors? If the protein is more abundant, the effect of the inhibitors should be less pronounced, with respect to the cells not expressing HMGA2. In addition, if the DLD1-Vector cells do not express HMGA2 at all, why the inhibitors are so effective?
Also results presented in fig. 3 A and B are questionable, since the effects of the inhibitors in very strong also in the cells not expressing HMGA2.
In conclusion, I do not think that the results presented in this paper are conclusive.
Moreover, even if English is not my mother language, I must say the quality of the language is very poor. It is not a matter of style: the English of this paper is so poor that sometime ,it is difficult to understand the message.